# The Growth Factors and Cytokines of Dental Pulp Mesenchymal Stem Cell Secretome May Potentially Aid in Oral Cancer Proliferation

**DOI:** 10.3390/molecules26185683

**Published:** 2021-09-19

**Authors:** A. Thirumal Raj, Supriya Kheur, Zohaib Khurshid, Mohammed E. Sayed, Maryam H. Mugri, Mazen A. Almasri, Manea Musa Al-Ahmari, Vikrant R. Patil, Shilpa Bhandi, Luca Testarelli, Shankargouda Patil

**Affiliations:** 1Department of Oral Pathology and Microbiology, Dr. D.Y. Patil Dental College and Hospital, Dr. D.Y. Patil Vidyapeeth, Pune 411018, India; thirumalraj666@gmail.com; 2Department of Prosthodontics and Dental Implantology, College of Dentistry, King Faisal University, Al-Ahsa 31982, Saudi Arabia; drzohaibkhurshid@gmail.com; 3Department of Prosthetic Dental Sciences, College of Dentistry, Jazan University, Jazan 45142, Saudi Arabia; drsayed203@gmail.com; 4Department of Maxillofacial Surgery and Diagnostic Sciences, College of Dentistry, Jazan University, Jazan 45142, Saudi Arabia; dr.mugri@gmail.com; 5Department of Oral and Maxillofacial Surgery, Faculty of Dentistry, King Abdulaziz University, Jeddah 21589, Saudi Arabia; malmasri@kau.edu.sa; 6Department of Periodontics and Community Dental Sciences, College of Dentistry, King Khalid University, Abha 62564, Saudi Arabia; abudanahmm@gmail.com; 7Biogenre Private Limited, Pune 412105, India; patilvikrant.r@gmail.com; 8Department of Restorative Dental Sciences, College of Dentistry, Jazan University, Jazan 45142, Saudi Arabia; shilpa.bhandi@gmail.com; 9Department of Oral and Maxillofacial Sciences, Sapienza University of Rome, 00185 Rome, Italy; luca.testarelli@uniroma1.it; 10Department of Maxillofacial Surgery and Diagnostic Sciences, Division of Oral Pathology, College of Dentistry, Jazan University, Jazan 45142, Saudi Arabia; dr.ravipatil@gmail.com

**Keywords:** conditioned media, dental pulp, mesenchymal stem cells, oral cancer, secretome

## Abstract

Background: Growth factors and cytokines responsible for the regenerative potential of the dental pulp mesenchymal stem cell secretome (DPMSC-S) are implicated in oral carcinogenesis. The impact and effects of these secretory factors on cancer cells must be understood in order to ensure their safe application in cancer patients. Objective: We aimed to quantify the growth factors and cytokines in DPMSC-S and assess their effect on oral cancer cell proliferation. Materials and methods: DPMSCs were isolated from patients with healthy teeth (*n* = 5) that were indicated for extraction for orthodontic reasons. The cells were characterized using flow cytometry and conditioned medium (DPMSC-CM) was prepared. DPMSC-CM was subjected to a bead-based array to quantify the growth factors and cytokines that may affect oral carcinogenesis. The effect of DPMSC-CM (20%, 50%, 100%) on the proliferation of oral cancer cells (AW123516) was evaluated using a Ki-67-based assay at 48 h. AW13516 cultured in the standard growth medium acted as the control. Results: VEGF, HCF, Ang-2, TGF-α, EPO, SCF, FGF, and PDGF-BB were the growth factors with the highest levels in the DPMSC-CM. The highest measured pro-inflammatory cytokine was TNF-α, followed by CXCL8. The most prevalent anti-inflammatory cytokine in the DPMSC-CM was IL-10, followed by TGF-β1 and IL-4. Concentrations of 50% and 100% DPMSC-CM inhibited Ki-67 expression in AW13516, although the effect was non-significant. Moreover, 20% DPMSC-CM significantly increased Ki-67 expression compared to the control. Conclusions: The increased Ki-67 expression of oral cancer cells in response to 20% DPMSC-CM indicates the potential for cancer progression. Further research is needed to identify their effects on other carcinogenic properties, including apoptosis, stemness, migration, invasion, adhesion, and therapeutic resistance.

## 1. Introduction

Mesenchymal stem cells derived from the dental pulp (DPMSC) possess a high capacity for differentiation and regeneration. They exhibit greater proliferation than other stem cells. The dental pulp is a readily accessible source of postnatal stem cells. Collectively, these properties facilitate their ex vivo expansion and make them an attractive choice of source for mesenchymal stem cells. Their clinical application in the maxillofacial region is the subject of extensive studies [1,2,3,4,5]. A major advantage of DPMSC is its ability to modulate the host’s immune system, preventing any significant adverse reaction [6]. Their inherent homing ability towards tissues with injury and pathology including cancers have led to an interest in their application as a drug delivery agent [7].

Cell-free therapy based on derived secretome/conditioned media has several advantages over cell-based therapy, including relative ease of processing and long-term storage. Cell-free derivatives have a lower risk of losing their functional properties [8]. As secretory factors (DPMSC-S) are the primary effectors of a DPMSC, the application of cell-free therapy is highly advocated [5,8].

In dentistry, DPMSC-S is utilized as a means to augment dental implant osseointegration [4]. Based on DPMSC-S’s anti-fibrotic properties, it may have application as a therapeutic agent in oral submucous fibrosis [9]. The regenerative properties of DPMSC may have wide applications in oral oncology. Given the significant morbidity associated with cancer-related surgical and therapeutic interventions, DPMSC-S could be used as a major rehabilitation agent for improving the quality of life by regenerating lost tissues and by ameliorating any post-treatment fibrosis.

Despite a wide array of potential applications, caution must be exercised in using DPMSC-S as they have been linked to the proliferation of prostate cancer cells [10,11]. DPMSCs secrete cytokines and growth factors that may have pro-carcinogenic effects. Considering the worrying preliminary evidence for DPMSC in promoting prostate cancer, our research sought to examine the effect of DPMSC on oral cancer cells (AW123516) and to delineate the DPMSC secretome profile.

## 2. Materials and Methods

### 2.1. DPMSC Isolation and Characterization

Ethical approval was obtained from the Institutional Ethics Review Board of the hospital where the patients were seen—Dr. D. Y. Patil Vidyapeeth, Pune, India (Reference number: D.Y.P.V/E.C/101/18). Five healthy patients of 18 to 21 years of age with sound teeth indicated for orthodontic extraction were enrolled. Written informed consent was obtained from all participants. Under aseptic conditions, the teeth were drilled and the pulp cavity was exposed. Pulp removal was carried out using a broach. The tissue was transported in a suitable basal medium (DMEM) to the laboratory. The pulp tissues were washed thrice using a phosphate-buffered saline (PBS) solution within a sterile Class II laminar flow hood. We employed a widely used method of explant culture as described by Patil et al. (2018) [12] to isolate and characterize the DPMSC. Tiny fragments were obtained by mincing the pulp tissues, which were then placed in 35 mm PS culture dishes. The tissues were covered completely with fetal bovine serum (FBS) (Gibco, MD, USA). The explanted tissue with FBS was incubated at a temperature of 37 °C and conditions of 5% CO_2_ for 24 h to ensure that the explants adhered to the surface of the flask. After 24 h, a complete growth medium (DMEM, Invitrogen, Carlsbad, CA, USA; combined with a 10% concentration of 10% FBS and an antibiotic–antimycotic solution) was used to maintain the DPSCs culture. Replenishment of the media was carried out twice every week. After 3 days, the DPSCs’ outgrowth started from the explants. An inverted phase-contrast microscope was used to closely observe the growth, morphology, and health of the cultured cells. A solution of 0.25% concentration of Trypsin–EDTA (Invitrogen, Carlsbad, CA, USA) was used to detach the cells at a confluence of 70–80%. The detached cells were then placed in a 25 sq. cm PS culture flask (Nunc, Rochester, NY, USA). At 1:2 ratio, the cells were continuously passaged, and 3 to 5 passaged cells were used in the current experiments.

After achieving a logarithmic phase, trypsinization was performed. The cells were harvested and washed twice using phosphate-buffered saline. The cells were fixed using 500 µL of a fixative solution. Their concentration was adjusted to 1 × 107 cells per milliliter. Then, 100 µL of the suspension was transferred to a fresh sterile Eppendorf tube. The cells were blocked using 50 µL of normal goat serum for 15 min at ambient room temperature. After blocking, the cells were incubated in the dark for 30 min with a fluorophore-tagged antibody for mesenchymal stem cell markers + (anti-CD90-PE, anti-CD73-FITC, anti-CD105-FITC), hematopoietic markers -ve (anti-CD34-APC, anti-CD45-APC), and Major Histocompatibility Complex Class II antigen (anti-HLA-DR-PE). After incubation, the cells were washed twice with PBS to remove any excess antibodies. The cells were analyzed and quantified using attune NxT (Thermo Fisher Scientific, Waltham, MA, USA). APC, FITC, and isotype control (PE-labeled immunoglobulin G-stained cells) acted as negative controls.

DPMSC-conditioned medium (DPMSC-CM) preparation: The dental pulp-derived mesenchymal stem cells were seeded into T-75 flasks having a surface area of 75 sq. cm (Nunc, Rochester, NY, USA). The cells were seeded at a density of 1 × 105 cells per sq. cm. A solution of DMEM -15 mL (Invitrogen, Carlsbad, California, USA) and a 10% concentration of FBS (Gibco, MD, USA) were used. The cells were incubated for twenty-four hours. After incubation, the contents of the flasks were viewed under a microscope to ensure cell adhesion. The media in the flasks were replaced with a solution of DMEM (15 mL) and a 0.2% concentration of FBS. The metabolic activity and cell morphology were observed to ensure that the cells did not undergo serum starvation. The cells were incubated for forty-eight hours. After incubation, the medium (DPSC-CM) was aspirated and centrifuged at 300 xg to separate any debris material for 5 min. The resultant supernatant was removed into fresh tubes and filtered into fresh sterile tubes using 0.22-micron syringe filters (Corning, NY, USA). The conditioned medium was stored at a temperature of −80 ºC until it was used in the experiment. For use in the experiment, the DPMSC conditioned medium was diluted with a 10% concentration of FBS. For our experiment, DPMSC-CM was produced as DPMSC-CM + 10% FBS. For experimental groups, 20%, 50%, and 100% of this composition were mixed with an appropriate amount of complete growth medium (DMEM + 10% FBS). Control groups were incubated with only a complete growth medium (DMEM + 10% FBS).

### 2.2. DPMSC-CM Profiling

Ang-2, EGF, EPO, bFGF, G-CSF, GM-CSF, HGF, M-CSF, PDGF-AA, PDGF-BB, SCF, TGF- α, and VEGF were assessed using the LEGENDplex™ Human Growth Factor Panel (13-plex; Biolegend, San Diego, CA, USA; Cat. No. 740180). Pro-inflammatory cytokines (IL-1β, TNF-α, IL-17A, IL-12p70, IFN-γ, CXCL10, CCL2, CXCL8) and anti-inflammatory cytokines (IL-2, IL-4, IL-6, IL-10, and TGF- β1) were assessed using the LEGENDplex™ Human Essential Immune Response Panel (13-plex; Biolegend, San Diego, CA, USA; Cat. No. 740929). Standardized protocols were followed as specified in the manufacturer’s instructions. First, 25 µL of DPMSC-CM was incubated with the microbeads for a period of two hours. After incubation, the detection antibodies were introduced and subjected to a further incubation of thirty minutes. Post-incubation, the samples were washed with a buffer solution. After washing, the samples were centrifuged for 5 min at 2000 rpm to remove the supernatant and collect the pellet. The collected pellet was resuspended in sheath fluid (200 μL). A flow cytometer (Attune NxT, Thermo Fisher Scientific, Waltham, MA, USA) was used to acquire the samples. LEGENDplex™ Data Analysis Software (BioLegend, San Diego, CA, USA) was used to analyze the data.

### 2.3. Oral Cancer Cell Lines Source and Culture in DPMSC-CM

The Advanced Centre for Treatment, Research, and Education in Cancer, Tata Memorial Centre, Mumbai, India donated AW13516, an oral squamous cell carcinoma cell line. AW13516 was cultured in different concentrations of DPMSC-CM (20%, 50%, and 100%) for 48 h. The control group consisted of a culture medium of DMEM supplemented with 10% fetal bovine serum, 100 U/mL streptomycin, and 100 U/mL penicillin.

### 2.4. Ki-67 Quantification by Flow Cytometry

The AW13516 subjected to the DPMSC-CM were subjected to PBS washing (twice), followed by 1500 rpm centrifugation for 5 min. After discarding the resulting supernatant, vortexing was carried out to resuspend the pellet. As vortexing was being carried out, 3 mL ethanol (70%) was added to the cells. After continuing vortexing for 30 s, incubation was carried out for 1 h at −20 °C. PBS was used to wash the cells (twice), followed by resuspension at 5 × 105 cells per mL. Anti-Ki-67-FITC (Biolegend) antibody was added to the cell suspension (100 µL) and incubation was carried out for 30 min at room temperature in the dark. PBS was used to wash the cells (twice) and they were subjected to resuspension in PBS (500 μL). The cells were acquired on a flow cytometer (Attune NxT, Thermo Fisher Scientific, Waltham, MA, USA) for analysis.

### 2.5. Statistical Analysis

The results are shown as the mean ± standard deviation of the experimental values for each sample. Each treatment group was evaluated individually against the control group. An unpaired t-test (two-tailed) was used to analyze the data on GraphPad Prism 8 software (GraphPad Software, La Jolla, CA, USA) for each cytokine. A value of *p* < 0.05 was considered significant and *p* < 0.01 was considered highly significant (* *p* < 0.05, ** *p* < 0.01, and ns not significant).

## 3. Results

### 3.1. DPMSC Characterization

The isolated DPMSC showed MSC-like morphological characteristics (Figure 1A) and were positive for mesenchymal stem cell markers CD90, CD73, and CD105, and negative for hematopoietic markers CD34, CD45-APC, and MHC class II antigen HLA-DR-PE (Figure 1B–D).

### 3.2. Quantification of the Growth Factors and Cytokines in DPMSC-CM

The DPMSC growth factors and cytokine profile are summarized in Figure 2 and Figure 3, respectively. The most prevalent growth factor in DPMSC-CM was VEGF, with 413.56 pg/mL, followed by HCF at 107.07 pg/mL, Ang-2 at 47.28 pg/mL, TGF-α at 32.52 pg/mL, EPO at 23.72 pg/mL, SCF at 18.43 pg/mL, FGF basic at 13.51 pg/mL, and PDGF-BB at 11.9 pg/mL. The remaining growth factors constituted less than 5 pg/mL. The most prevalent pro-inflammatory cytokine was TNF-α at 14.16 pg/mL, followed by CXCL8 at 6.95 pg/mL. The remainder of the pro-inflammatory cytokines amounted to less than 5 pg/mL. The most prevalent anti-inflammatory cytokine was IL-10 at 10.37 pg/mL, followed by TGF-β1 at 8.1 pg/mL, and IL-4 at 5.27 pg/mL. The remaining anti-inflammatory cytokine amounted to less than 5 pg/mL.

### 3.3. Effect of the DPMSC-CM on the AW13516

The Ki-67 assay showed that DPMSC-CM at a 20% concentration caused a significant increase in Ki-67 expression compared to the control. At higher concentrations of 50% and 100%, Ki-67 expression was attenuated significantly compared to 20% DPMSC-CM. However, the attenuation was not statistically significant compared to the control (Figure 4).

## 4. Discussion

In recent years, stem cell research has grown exponentially. Dental pulp mesenchymal stem cells (DPMSCs) are multipotent cells extracted from dental tissue, making them a safe and easily accessible source. DPMSCs have been attracting interest owing to their potential for multipotent differentiation. They may have applications in stem-cell-based therapy in regenerative medicine and dentistry [4].

Growth factors and cytokine profiles of DPMSCs can change with age. Bhandi et al. [13] conducted an age-based analysis of the DPMSC-S. DPMSC isolated from a younger individual (14–28 years old) showed a greater amount of growth factors than DPMSC isolated from older individuals (50–75 years old). The DPMSC from the older individuals carried a higher level of pro-inflammatory cytokines than younger DPMSCs. The study found that growth factors deplete with age, while pro-inflammatory cytokines increase with age. This can cause an overall reduction in the regenerative potential of the DPMSC. Apart from age, the cell culture conditions of the DPMSC affect its secretome profile and regenerative potential. In a recent study by Bhandi et al. [14], the researchers were able to modulate the DPMSC secretome profile by induction of hypoxia through cobalt-chromium treatment to enhance its regenerative potential. Based on these observations, we hypothesized that the overall effect of the DPMSC relies on the levels of the growth factors and cytokine profile, which could be modulated to suit our needs.

Cancer patients undergoing surgical resection, radiotherapy, or chemotherapy often develop post-therapeutic morbidities that reduce their overall quality of life [15,16]. The use of DPMSC-S for regeneration could aid their rehabilitation.

Despite the promising applications of DPMSC-S in oral oncology, caution must be exercised in their deployment. A recent study by Dogan et al. revealed that DPMSC-S can augment the proliferation of prostate cancer cells [10,11]. The effect was attributed to the secretory factors. However, the study did not delineate the factors in the secretome that may have aided in cancer cell proliferation. Our study assessed whether the DPMSC-S can induce proliferation in oral cancer cells. Additionally, the DPMSC-S profile was delineated and quantified to indicate the potential factors that may affect cancer cells.

Our results, based on a Ki-67 assay, indicated that, at low concentrations, DPMSC-CM augmented proliferation. At higher concentrations, proliferation was inhibited. The bead-based array revealed that several growth factors (VEGF, HCF, Ang-2, TGF-α, EPO, SCF, FGF, and PDGF-BB) and pro-inflammatory cytokines (TNF-α and CXCL8) could be major contributing factors in the propagation of oral cancer cells owing to their higher levels compared to other factors in the secretome.

There have been few studies that have assessed the effects of stem cells isolated from dental tissue on cancer. A 2010 study by Rizvanov et al. [17] found that human third molar tooth germ stem cells (TGMSCs) rendered protection to SH-SY5Y cells (neuroblastoma cell line) against hydrogen peroxide-induced stress. The authors suggested that the neuroprotective effect stemmed from the secretome of the TGMSCs. Yalvac et al. in 2011 [18] found a similar protective effect induced by TGMSCs-CM in SH-SY5Y against doxorubicin/hydrogen peroxide-induced neurotoxicity. In 2013, Yalvac et al. [19] delineated the secretome profile of TGSC-CM and identified key factors, including growth factors TGF-b1, FGF2, VEGF, and cytokines IL-5, IL-6, IL-8, and IL-10, that were attributed as the primary effectors on the cancer cells.

As most studies of DPMSCs are based on in vitro culture models, it is unclear whether the effect of the cancer cells could translate in vivo. The same limitations apply to the present in vitro study, wherein there is a lack of immunogenic components and a local microenvironment. Based on an analysis of the current literature and research, it is clear that DPMSCs will have many therapeutic properties and a vital role in regenerative therapy. This may allow the treatment of hitherto untreatable conditions. However, additional research is imperative before their deployment. Further studies using in vivo models could clarify the role of the host immune components in modulating the effect of DPMSC on cancer cells and vice versa. Future research could explore modulating the secretory profiles of the DPMSCs to attenuate the pro-inflammatory cytokines in order to reduce the pro-carcinogenic effect.

## 5. Conclusions

In conclusion, based on the results of the present study, DPMSC-CM can proliferate oral cancer cells at low concentrations and attenuate proliferation at higher concentrations. These effects are largely mediated by the secretome profile of the DPMSCs. The factors responsible for cancer cell proliferation may be a higher level of pro-inflammatory cytokines (TNF-α and CXCL8) and growth factors (VEGF, HCF, Ang-2, TGF-α, EPO, SCF, FGF, and PDGF-BB). Future studies exploring the effect of the DPMSC-CM on other properties of cancer cells, including invasion, migration, therapeutic resistance, adhesion, and stemness, would further clarify and expand our knowledge base on stem cell interactions. In addition to oral cancer cell lines, the interaction of DPMSCs with cancers of other tissues could be examined to assess any differential effect. The future of stem-cell-based therapies will depend on the understanding of their biology and their interactions with other cells.

## Figures and Tables

**Figure 1 molecules-26-05683-f001:**
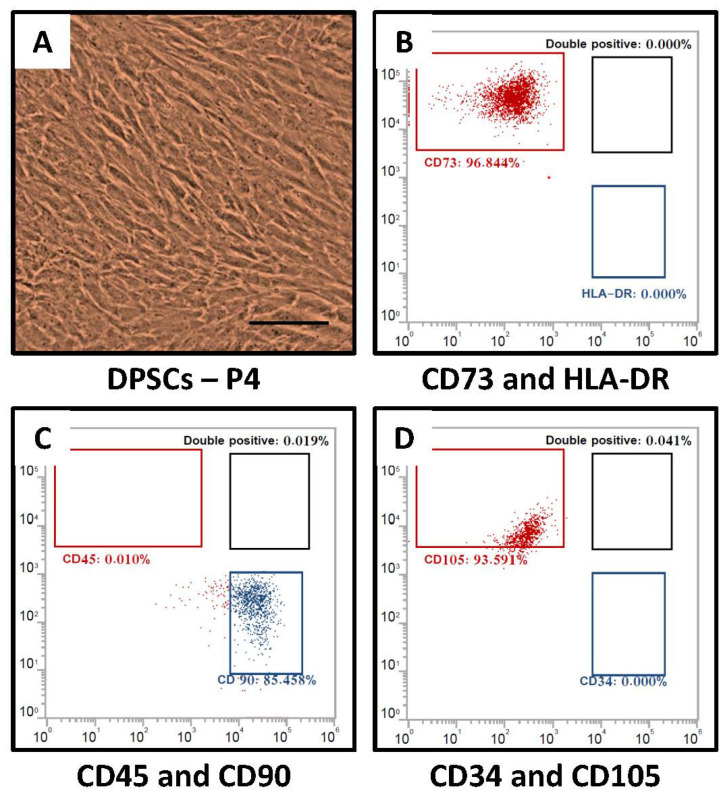
Characterization of DPSCs for mesenchymal stem cell properties. (**A**) Photomicrograph of DPSCs from one of the five donors at passage 2. Scale bar = 100 μm, (**B**–**D**) DPSCs were checked for MSC-specific positive markers CD73, CD90, and CD105 and MSC-specific negative markers CD34, CD45, and HLA-DR. DPSCs-P4: Dental pulp stem cells passage 4.

**Figure 2 molecules-26-05683-f002:**
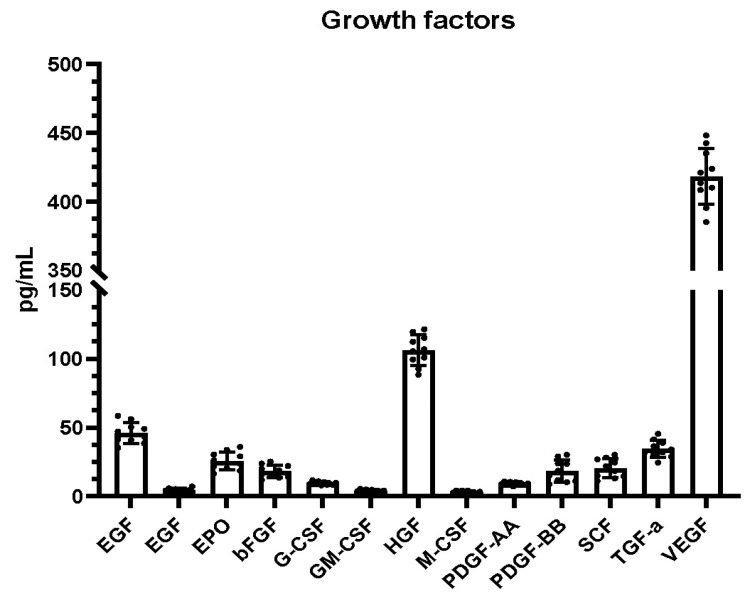
DPMSC-CM growth factor profile after 48 h incubation with DMEM + 0.2% FBS. VEGF: Vascular endothelial growth factor, TGF-α: Transforming growth factor-alpha, SCF: Stem cell factor, PDGF-AA: Platelet-derived growth factor AA, PDGF-BB: Platelet-derived growth factor-BB, HGF: Hepatocyte growth factor, FGF basic: Basic fibroblast growth factor, EPO: Erythropoietin, EGF: Epidermal growth factor, Ang-2: Angiopoietin-2, G-CSF: Granulocyte colony-stimulating factor, GM-CSF: Granulocyte-macrophage colony-stimulating factor, M-CSF: Macrophage colony-stimulating factor.

**Figure 3 molecules-26-05683-f003:**
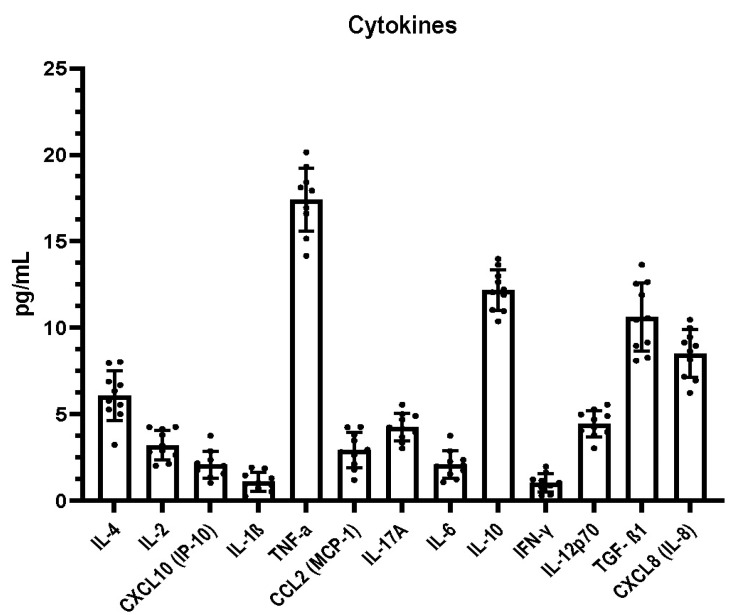
DPMSC-CM cytokine profile after 48 h incubation with DMEM + 0.2% FBS (*n* = 5). IL-4: Interleukin 4, IL-2: Interleukin 2, CXCL10: C-X-C motif chemokine ligand 10, IL-1β: Interleukin 1 beta, TNF-α: Tumor necrosis factor alpha, CCL2: C-C motif chemokine ligand 2, IL-17A: Interleukin 17A, IL-6: Interleukin 6, IL-10: Interleukin 10, IFN-γ: Interferon gamma, IL-12p70: Interleukin 12, CXCL8: Interleukin 8, TGF-β1: Transforming growth factor beta 1.

**Figure 4 molecules-26-05683-f004:**
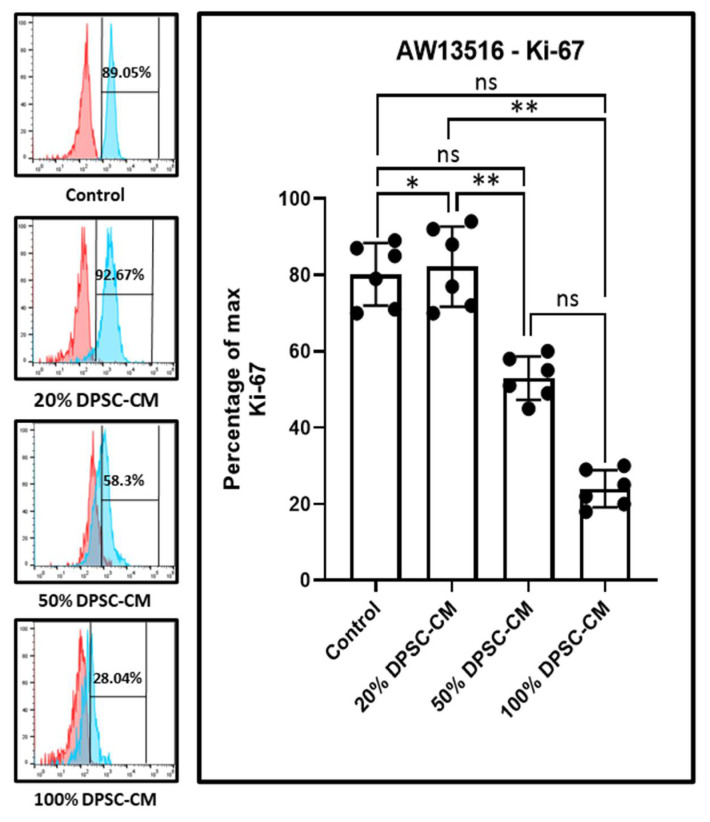
Dose-dependent effect of the DPMSC-CM (*n* = 3) on the Ki-67 status of AW13516 at 48 h. * *p* < 0.05, ** *p* < 0.01, ns not significant. DPSC-CM: Dental pulp stem cell-conditioned medium, Ki-67: Marker of proliferation Ki-67.

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
