# Peer review of "The Growth Factors and Cytokines of Dental Pulp Mesenchymal Stem Cell Secretome May Potentially Aid in Oral Cancer Proliferation"

_molecules, 2021, doi:10.3390/molecules26185683_

Round 1
Reviewer 1 Report
English was improved, though some linguistic errors along with syntactical mistakes are still present and need to be corrected.
The other suggestions were addressed by the authors.
Author Response
Academic Editor comments:
The authors should state that Figure 1 was obtained from DPMSCs derived from donor "X" and is representative of the five donors. Or explain what exactly is being presented in this figure. Were samples from all 5 donors united into one culture?
Response: The photomicrograph is representative of DPSCs from just 1 donor. It is shown just to depict the morphological characteristics of DPSCs under a microscope. Other donor DPSCs were cultured independently.
The author's should explain why 100%, 50%, and 20% dilutions were selected and address the possibility that poor growth in the more concentrated conditioned media results from depletion of nutrients or the presence of additional factors derived from the DPMSCs that were not examined.
Response: We wanted to see if the DPSC-CM affects the growth of the cancer cells in a concentration dependent manner. But there is no standard data available on the concentration gradient of the CM. Complete growth medium (DMEM + 10% FBS) was used for dilution, but before that 10% FBS was also mixed with CM used for the treatment. We have kept control group treated with complete growth medium. We have tried to keep the media composition uniform throughout the study (DMEM/CM + 10% FBS). Initially, it was our objective to see if multiple growth factors and cytokines and other soluble factors secreted by DPSCs can synergistically affect the homeostasis of the cancer cells. In our future studies we are aiming to perform high-throughput analyses including proteomics and metabolomics to decipher these mechanisms for each individual factors. Furthermore, we will employ in silico analysis to decode the exact signalling pathways involved. Lately, mesenchymal stem cells including DPSCs are in the limelight for their regenerative applications. Apart from that, these cells are being explored thoroughly as alternative therapies in many types of cancers. Transplantation of other types of MSCs such as bone marrow derived MSCs and adipose tissue derived MSCs have shown suppressive effect on many types of tumors. The ability of MSCs to suppress the tumors underlies in their paracrine secretions upon transplantation as investigated extensively. Our study is a very first time report of this kind showing differential effects of paracrine secretions of DPSCs on oral cancer cell line. In this study, we have expressed our opinion on whether these results tend to be pro-tumorigenic or anti-tumorigenic.
Reviewer 1:
English was improved, though some linguistic errors along with syntactical mistakes are still present and need to be corrected.
The other suggestions were addressed by the authors.
Reply; As advised, the linguistic errors along with syntactical mistakes in the manuscript were corrected
Reviewer 2:
The authors have modified the paper according to the referee suggestions, I believe that it can be accepted for publication
Reply: The authors thank the reviewer for the comments
Reviewer 2 Report
The authors have modified the paper according to the referee suggestions, I believe that it can be accepted for publication
Author Response
Academic Editor comments:
The authors should state that Figure 1 was obtained from DPMSCs derived from donor "X" and is representative of the five donors. Or explain what exactly is being presented in this figure. Were samples from all 5 donors united into one culture?
Response: The photomicrograph is representative of DPSCs from just 1 donor. It is shown just to depict the morphological characteristics of DPSCs under a microscope. Other donor DPSCs were cultured independently.
The author's should explain why 100%, 50%, and 20% dilutions were selected and address the possibility that poor growth in the more concentrated conditioned media results from depletion of nutrients or the presence of additional factors derived from the DPMSCs that were not examined.
Response: We wanted to see if the DPSC-CM affects the growth of the cancer cells in a concentration dependent manner. But there is no standard data available on the concentration gradient of the CM. Complete growth medium (DMEM + 10% FBS) was used for dilution, but before that 10% FBS was also mixed with CM used for the treatment. We have kept control group treated with complete growth medium. We have tried to keep the media composition uniform throughout the study (DMEM/CM + 10% FBS). Initially, it was our objective to see if multiple growth factors and cytokines and other soluble factors secreted by DPSCs can synergistically affect the homeostasis of the cancer cells. In our future studies we are aiming to perform high-throughput analyses including proteomics and metabolomics to decipher these mechanisms for each individual factors. Furthermore, we will employ in silico analysis to decode the exact signalling pathways involved. Lately, mesenchymal stem cells including DPSCs are in the limelight for their regenerative applications. Apart from that, these cells are being explored thoroughly as alternative therapies in many types of cancers. Transplantation of other types of MSCs such as bone marrow derived MSCs and adipose tissue derived MSCs have shown suppressive effect on many types of tumors. The ability of MSCs to suppress the tumors underlies in their paracrine secretions upon transplantation as investigated extensively. Our study is a very first time report of this kind showing differential effects of paracrine secretions of DPSCs on oral cancer cell line. In this study, we have expressed our opinion on whether these results tend to be pro-tumorigenic or anti-tumorigenic.
Reviewer 1:
English was improved, though some linguistic errors along with syntactical mistakes are still present and need to be corrected.
The other suggestions were addressed by the authors.
Reply; As advised, the linguistic errors along with syntactical mistakes in the manuscript were corrected
Reviewer 2:
The authors have modified the paper according to the referee suggestions, I believe that it can be accepted for publication
Reply: The authors thank the reviewer for the comments
This manuscript is a resubmission of an earlier submission. The following is a list of the peer review reports and author responses from that submission.
Round 1
Reviewer 1 Report
This study quantified the amount of representative cytokines and growth factors contained in DPSC-CM. Furthermore, the effect of DPSC-CM on the growth of cancer cells was investigated. The undiluted and 50% diluted DPSC-CM suppressed the growth of cancer cells, while the 20% DPSC-CM slightly promoted the cell growth. The authors conclude that DPSC-CM promotes the growth of cancer cells.
This study has many problems and should be rejected.
1. Fig1 regend states that DPSC was cultivated in complete medium to produce DPSC-CM, but no explanation was given for "complete medium". Does DPSC-CM contain bovine serum? There is no description how the DPSC-CM was manufactured in the experimental method.
2. Although it is stated that DPSC was collected from a five donors, the measurement results of cytokine and growth factor levels have not been statistically analyzed in Fig 1and 2. Did the author measure the cytokine levels of the five DPSC-CMs? The author must show the reproducibility of the measurmemnts. Measure at least three times for five DPSC-CMs.
3. Why did you use DPSC-CM without dilution, 50% dilution, and 20% dilution? What is the meaning of this dilution? What medium was used for dilution? If the dilution medium containing FBS, how did they affect the cancer cell's growth?
4. The reason why the undiluted DPSC-CM suppressed the growth of cancer cells may be an exhaustion of the medium by the DPSC culture. DPSC-CM contains a large amount of cell waste products.
Reviewer 2 Report
This manuscript in its actual state, it is not publishable in this journal or other relevant journals in the field. The english needs extensive revision. I advise that the authors should request help for this.
In the introduction section, there are some sentences that are not scientifically correct. At the end of the first paragraph, reference 6 is not concerned to DPMSCs but to BMSCs, therefore what is mentioned is not true. In line 60, the sentence is not referenced, it should have a reference showing this. If there is not, the authors cannot state this.
At the beginning of the second paragraph, the authors wrote that “despite DPMSC safety”, and this is not evidenced still. The last sentence of this paragraph is also very awkward: “Also, as the primary effectors of a DPMSC are their secretory factors (DPMSC-S), the emphasis for a cell-free therapy is not only based on caution but also efficacy [5].”
In the MM section, there are several errors. Starting with the culture of the explants, the authors wrote in line 92 that the tissue was cultured WITH 20%FBS, based on a previous publication that says that it was DMEM with 20% FBS. The last sentence of this paragraph: “At 1:2 ratio, the cells were continuously passed and the passage 3 to 5 cells were used in the current experiments” is not perceptible.
In line 111, it should be mentioned that the instrument used was a flow cytometer. Because when they describe the Ki-67 quantification they wrote flow cytometer, but do not specify which one. Concerning this, the title should be Ki-67 quantification by flow cytometry.
In the results section, the first results are not shown, this should be mentioned.
Figure legends are above the figures, which is not correct. Also, the figure captions are too short and do not evidence enough information to understand the figures. No statistics was mentioned in MM section, but in figure 3 it is present some type of elements that might be statistical analysis although not referred in the figure caption. The authors do not describe how many repetitions were performed or replicates. This is not acceptable.
The discussion and conclusion sections are in line to what was previously described. Bad English and general conclusions that no not evidence anything to the state of the art in the field.
Reviewer 3 Report
This paper studied the growth factors (VEGF, HCF, Ang-2, TGF-α, EPO, SCF, FGF, and PDGF) and cytokines (TNF-α and CXCL8) in DPMSC-S and assess its effect on the oral cancer cell proliferation.
The results are speculative as obtained in vitro, the authors should enlarge in the discussion and conclusion the limitation of this study.
They should also discuss the results on the differences on MIB-1 expression.